# GRAPH SPLINE NETWORKS FOR EFFICIENT CONTINUOUS SIMULATION OF DYNAMICAL SYSTEMS

## ABSTRACT

While complex simulations of physical systems have been widely studied in engineering and scientific computing, lowering their often prohibitive computational requirements has only recently been tackled by deep learning approaches. In this paper, we present GRAPHSPLINENETS, a novel deep learning approach to speed up simulation of physical systems with spatio-temporal continuous outputs by exploiting the synergy between *graph neural networks* (GNN) and *orthogonal spline collocation* (OSC). Two differentiable OSC (time-oriented OSC and spatial-oriented OSC) are applied to bridge the gap between discrete GNN outputs and generate continuous solutions at any location in space and time without explicit prior knowledge of underlying differential equations. Moreover, we introduce an adaptive collocation strategy in space to enable the model to sample from the most important regions. Our model improves on widely used graph neural networks for physics simulation on both efficiency and solution accuracy. We demonstrate GRAPHSPLINENETS in predicting complex dynamical systems such as the heat equation, damped wave propagation and the Navier-Stokes equations for incompressible flows, where they improve accuracy of more than $25\%$ while providing at least $60\%$ speedup.

## 1 INTRODUCTION

For a growing variety of fields, simulations of *partial differential equations* (PDEs) representing physical processes are an essential tool. PDE–based simulators have been widely employed in a range of practical issues, spanning from astrophysics (Mücke et al., 2000) to biology (Quarteroni & Veneziani, 2003), engineering (Wu & Porté-Agel, 2011), finance, (Marriott et al., 2015) or weather forecasting (Bauer et al., 2015). Traditional solvers for phsysics-based simulation oftentimes need a significant amount of computational resources (Houska et al., 2012), such as solvers based on first principles and the modified Gauss-Newton methods. To broaden the scope of applications of dynamics simulation, the scientific machine learning community has put considerable effort into developing computationally simple yet accurate simulation approaches.

Deep learning has been shown to be a powerful alternative to efficiently compute solutions (Raissi et al., 2019) or model dynamical systems directly from data (Mrowca et al., 2018). Among deep learning methods, *graph neural networks* (GNNs) come with desirable properties such as spatial equivariance and translational invariance which allow learning representations of dynamical interactions in a generalizable manner (Pfaff et al., 2021; Bronstein et al., 2021) and on unstructured grids. Despite the benefits of these paradigms, graph-based models have the fundamental drawback of being discrete in nature, which makes it challenging to implement continuous simulations in time and space. While graph models that operate in continuous space or continuous time have been introduced in the past (Poli et al., 2019), such approaches mainly deal with only one aspect of continuity at once, either in space or time, and are hindered by accuracy issues while interpolating in space (Alet et al., 2019) or require a considerable number of iterative evaluations of a vector field in time limiting their performance (Xhonneux et al., 2020).

To bridge the gap between the inherently *discrete* graphs and the intrinsic *continuous* nature of the real world, in this work we propose GRAPHSPLINENETS, a novel method that exploits the synergy between graph neural networks and the *orthogonal spline collocation* (OSC) method (Bialecki & Fairweather, 2001; Fairweather & Meade, 2020). By leveraging the OSC, our approach can

produce predictions at any location in space and time without explicit prior knowledge of the underlying differential equation. GRAPHSPLINENETS achieve significant speedups compared to GNN baselines by making use of efficient sparse linear solvers (de Boor & Weiss, 1980) for the OSC problem and training the model end-to-end with larger temporal resolutions. Moreover, thanks to the *super-convergent* approximations at nodes of the OSC partition (Qiao et al., 2021) and an adaptive sampling strategy of collocation points, GRAPHSPLINENETS improve the solution accuracy of predictions in continuous space and time.

We summarize our contributions as follows:

- We introduce GRAPHSPLINENETS, a learning framework for complex dynamical system in continuous time and space leveraging the OSC method.

- We introduce an adaptive collocation sampling strategy to improve accuracy and a differentiable algorithm for fast inference of the OSC that allows for end-to-end training.

- We demonstrate that GRAPHSPLINENETS outperform or are competitive against baselines in predicting continuous complex dynamics in terms of both accuracy and speed.

## 2    RELATED WORKS

**Graph Neural Networks for Dynamics Predictions**    Deep neural networks have recently been successfully employed in a variety of different tasks, ranging from simulated (Long et al., 2018; Li et al., 2020) and real datasets (Pathak et al., 2022; Li et al., 2022a; Poli et al., 2022) demonstrating their capabilities in predicting complex dynamics often orders of magnitude faster than traditional numerical solvers. We aim at finding efficient and accurate surrogate models: unlike data-driven approaches for solving PDEs such as PINNs (Raissi et al., 2019), that aim at finding solutions to a set of equations, our methods does not need to know the exact equations of a dynamical system and can directly learn mappings from data. One major line of work for dynamics prediction involves the use of graph neural networks (GNNs): these models provide several benefits compared to other deep learning methods based on regular grids such as convolutional networks. In particular, they make it possible to learn on irregular grids and varying connectivity and inherit physical properties derived from geometric deep learning, such as permutation and spatial equivariance (Bronstein et al., 2021). Alet et al. (2019) represent adaptively sampled points in a graph architecture to simulate continuous underlying physical processes without any *a priori* graph structure. Sanchez-Gonzalez et al. (2020) introduce particle-based graph simulators with dynamically changing connectivity simulating interactions through message passing; Pfaff et al. (2021) extend particle-based simulations to mesh-based ones. GNN-based approaches have also been shown to represent some parts of classical numerical solvers, such as finite differences and volumes (Brandstetter et al., 2022; Lienen & Günnemann, 2022). Graph neural networks have also recently been applied to large-scale weather predictions (Keisler, 2022). Another deep learning direction on irregular grids that avoids graphs altogether is to convert the input domain into a regular grid via learnable deformations to make usage of neural operators possible (Li et al., 2022b). Compared to other deep learning methods for physics predictions on unstructured grids, we do not need to learn transforms and predict directly in the target domains; moreover, we place emphasis on bridging the gap between discrete graph nodes in space and time by allowing for fast and accurate continuous predictions.

**Collocation Methods and Graphs**    Collocation and interpolation methods[1] are used to estimate unknown data values from known ones (Bourke, 1999). GNNs for predicting dynamics inherently lack an important aspect characterizing physical systems: continuity. The concept of continuity can be separated into two categories: *continuity in time* and *continuity in space*. The former has been investigated by using continuous ODE models (Poli et al., 2019; Xhonneux et al., 2020; Fang et al., 2021) that can, in theory, represent a system evolving in time continuously. However, such methods employ numerical solvers that introduce a considerable number of function evaluations and do not consider the fact that deep learning models can be capable of overcoming time discretization

---

[1]Collocation and interpolation are terms that are oftentimes used interchangeably. While interpolation is defined as obtaining unknown values from known ones, collocation is usually defined as a finite solution space satisfying equations dictated by known (collocation) points. Thus, collocation can be considered as a flexible subset of interpolation methods that satisfies certain conditions, such as $\mathcal{C}^1$ class continuity.

errors by learning residual values (Poli et al., 2020; Berto et al., 2022). In our approach, we aim at learning state updates directly rather than vector fields to allow for fast predictions. In the case of space continuity, the graph structure itself is inherently discrete. Alet et al. (2019) employ linear interpolation in graph learning; however, this interpolation method is known to be inaccurate and does not respect $\mathcal{C}^1$ continuity. Unlike earlier methods, we employ the *orthogonal spline collocation* (OSC) (Bialecki & Fairweather, 2001) method to quickly find $\mathcal{C}^1$ continuous solutions to differential equations given a small number of partition points, producing a space-time continuous simulation. Unlike methods requiring full matrix inversion, OSC has a complexity of only $O(n^2 \log n)$ thanks to its sparse structure, allowing for fast inference. Another strain of work involves directly learning collocation weights (Guo et al., 2019; Brink et al., 2021); in our case, however, we use graphs to predict spatio-temporal locations of nodes and employ the OSC approach in an end-to-end manner. A further benefit of the OSC choice are its theoretical guarantees on convergence (Bialecki, 1998).

# 3 SPLINEGRAPHNETS

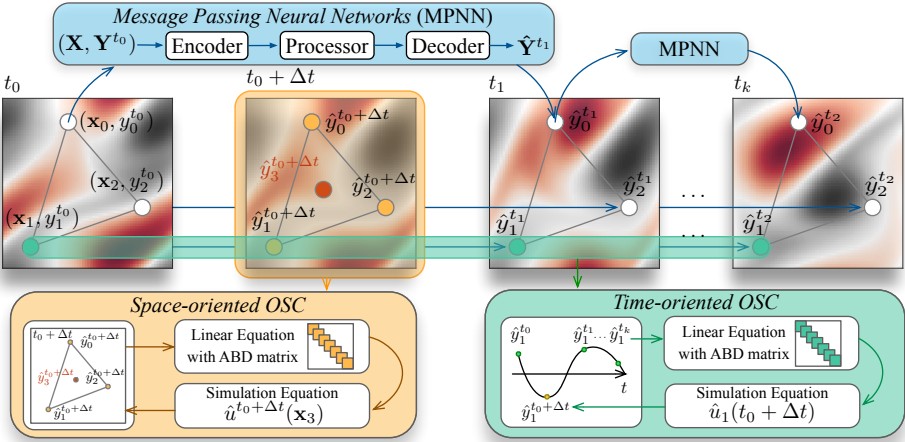

Figure 1: The overall scheme of GRAPHSPLINENETS.

## 3.1 PROBLEM SETUP

We first introduce necessary notation that we will use throughout the paper. Let superscripts denote time and subscripts denote space indexes as well as others with an abuse of notation. We denote the state of a PDEs process at physical space location $\mathbf{X} = \{\mathbf{x}_i, i = 1, \cdots, N\}$ at time $t$ as $\mathbf{Y}^t = \{y_i^t, i = 1, \cdots, N\}$ where $N$ represents the number of sample points and $\Omega$ is the physical domain. More specifically, we have $\mathbf{x}_i \in \Omega \subset \mathbb{R}^D, y_i^t \in \mathbb{R}$ can be described by a solution of the PDEs, i.e. $y_i^t = u(\mathbf{x}_i, t)$. The final objective of a physics process simulator is to estimate future states $\mathbf{Y}^{T+1}$ given the history states $\{\hat{\mathbf{Y}}^i, i = 1, \cdots, T\}$. Long-term predictions can be obtained via autoregressive rollouts of the model. In our method, we also aim to infer a spatio-temporal continuous prediction $\hat{\mathbf{Y}}^{T+\Delta t} = \hat{u}^{T+\Delta t}(\mathbf{x}), \mathbf{x} \in \Omega, \Delta t \in \mathbb{R}_+$, where the $\hat{u}^{T+\Delta t}$ is the simulated functions of the domain at time $T + \Delta t$.

## 3.2 METHOD OVERVIEW

Fig. 1 depicts the entire architecture of our model that can be divided into three main components: message passing neural networks (MPNN), space–oriented collocation and time–oriented collocation. The MPNN takes history state observations as input and infers a sequence of discrete future predictions via autoregressive rollouts. On these discrete predictions, we then use the time–oriented and space–oriented collocation methods to obtain simulation functions that can provide both time and space continuous simulations.

### 3.3 MESSAGE PASSING NEURAL NETWORKS

We employ a graph $\mathbf{G}^t = \{\mathbf{V}^t, \mathbf{E}^t, \} \in \mathcal{G}$ to represent the nodes in the physics domain, where $\mathbf{V}^t = \{\mathbf{v}_i^t\}_{i=0}^N$ and $\mathbf{E}^t = \{\mathbf{e}_{ij}^t\}_{i,j=0}^N$. $\mathbf{v}_i^t$ and $\mathbf{e}_{ij}^t$ denote the attribute of sample node $i$ and the attribute of the directed edge between node $i$ and $j$, respectively. Each node has the attribute encoded from its state information. The MPNN employs an encoder–processor–decoder structure to predict the states of sample points at the next timestep:

$$\hat{\mathbf{Y}}^{t+1} = \underbrace{\mathcal{D}}_{\text{decoder}} \left( \underbrace{\mathcal{P}_m(\cdots(\mathcal{P}_1(}_{\text{processor}}\underbrace{\mathcal{E}(\mathbf{X}, \mathbf{Y}^t)}_{\text{encoder}}))) \right) \tag{1}$$

where $\mathcal{E}(\cdot)$ is the encoder, $\mathcal{P}_i(\cdot)$ is the $i$–th message passing layer, and $\mathcal{D}(\cdot)$ is the decoder.

### 3.4 ORTHOGONAL SPLINE COLLOCATION METHOD

The *orthogonal spline collocation* (OSC) method consists of four steps in total: (1) partitioning and selection of collocation points (2) generating the equation of simulator polynomial parameters (3) solving equations and (4) simulating in the physical domain. In the following part of this section, we introduce the OSC method for the 1–D and 2–D cases, as both are relevant to the proposed model. GRAPHSPLINENETS continuously predicts in time domain using the 1–D OSC and in space domain using its 2–D counterpart.

**Time–oriented OSC** For a process of a specific sample point $\mathbf{x}_i$ in the physics domain with its state changing over time, we can consider it as an ordinary differential equation (ODE) process $f(u_i(t)) = 0, t \in [0, T]$, where $u_i(\cdot)$ is the solution with boundary conditions $u_i(0) = y_i^0, u(T) = y_i^T$. The target of time–oriented OSC is to find a series of polynomials under order $r$ and satisfy $C^1$ continuity to simulate the solution.

To find these polynomials, we select $N_p$ partitions in the time domain $\pi : 0 = t_{\text{p},0} < t_{\text{p},1} < \cdots < t_{\text{p},N_p} = T$. Note that these partitions can be not isometric. Then, we initialize one polynomial of order $r$ in each partition. These polynomials have $N_p(r-1)$ degrees of freedom in total, which is the number of variables to be specified to uniquely determine these polynomials. To decide these variables, we need to select $r-1$ collocation points in each partition; in total $N_c = N_p(r-1)$. In our model, we consider each message passing layers prediction time step as a collocation point $\{t_{\text{c},k}\}_{k=0}^{N_c}$, which means each $r-1$ rollout prediction belongs to one partition, i.e. $t_{\text{p,n}} < t_{\text{c,n}(r-1)} < \cdots < t_{\text{c},(n+1)(r-1)} < t_{\text{p,n}+1}, n = 0, \cdots, N_p - 1$.

By substituting the state of these collocation points $\{y_i^{t_k}\}_{k=0}^{N_c}$ to polynomials, we can transfer this simulation problem to an algebraic equation. We emphasize that the coefficient matrix of this algebraic equation is *almost block diagonal* (ABD) (De Boor & De Boor, 1978). This kind of system allows for efficient computational routines (Amodio et al., 2000), that we will introduce in § 3.5. By solving the equation, we obtain the simulation polynomial $\hat{u}_i(\cdot)$ that can be used to simulate the value for this sample point at any time $\hat{y}_i^{\Delta t}, \Delta t \in [0, T]$. All these steps can be calculated by matrix operations, so that the model is fully end-to-end differentiable.

**Space–oriented OSC** For a specific time frame $t_k$, the states of sample points in the physical domain $\{y_i^{t_k}\}_{i=0}^N$ can be described by PDEs $f(u^{t_k}(\mathbf{x}_i)) = 0, x_i \in \Omega$, where $u^{t_k}(\cdot)$ is the solution for the state at this time frame. The target of the space–oriented OSC is to find one polynomial of order $r$ on each partition for every dimension and make these $N \times D$ polynomials $C^1$ continuous in the domain.

Similar with the time–oriented OSC, we select $N_{\text{p},d}$ partitions in the $d$th dimension $\pi_d : B_{\inf,d} = p_{d,0} < p_{d,1} < \cdots < p_{d,N_{\text{p},d}} = B_{\sup,d}, d = 1, \cdots, D$, where $B_{\inf,d}, B_{\sup,d}$ are the lower and upper boundary of dimension $d$. Note that these partitions can be not isometric. Then we initialize one polynomial under order $r$ in each partition for each dimension. These polynomials have the degree of freedom $N_p^D(r-1)^D$ in total. To determine these polynomials, we need to select $r-1$ collocation points in each partition, in total $N_c = N_p^D(r-1)^D$. In our model, we consider the prediction states of sample points as collocation points $\{\mathbf{x}_{i,k}\}_{k=0}^{N_c}$.

By substituting the state of these collocation points $\{y_i^{t_k}\}_{k=0}^{N_c}$ to polynomials, we can transfer this simulation problem to an algebraic problem with an ABD coefficient matrix. By solving this problem we can obtain the simulation polynomial $\hat{u}^{t_k}$, that can be used for simulating the result in the whole domain $\hat{y}^{t_k} = \hat{u}^{t_k}(\mathbf{x}), \forall \mathbf{x} \in \Omega$. More details about deriving the degree of freedom, visualization of the ABD coefficient matrix, and further techniques of applying OSC are shown in Appendix A.

### 3.5 Algorithm for efficiently solving the ABD matrix

Most interpolation methods need to solve linear equations. Gaussian elimination is one of the most widely used methods to solve a (dense) linear equation, which however has a $O(n^3)$ complexity (Strassen et al., 1969). Even with the best algorithms known to date, the lower bound of time complexity to solve such equations is $O(n^2\log(n))$ (Golub & Van Loan, 2013). In the OSC method, the coefficient matrix for the linear equation follows the ABD structure, which we can efficiently solve with a time complexity $O(n^2)$ by the COLROW algorithm (Diaz et al., 1983) as shown in Fig. 2. The core idea for this method is that by using the pivotal strategy and elimination multipliers, we can decompose the coefficient matrix into a set of permutation matrix and upper or lower triangular matrix that can be solved in $O(n^2)$ time each. The

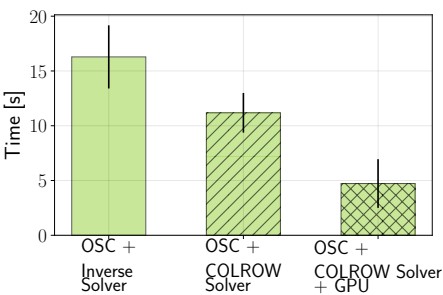

Figure 2: The COLROW algorithm allows for fast solutions to the OSC problem. GPU implementation further improves on inference times.

most recent package providing this algorithm is in the FORTRAN programming language: our re-implementation in PyTorch (Paszke et al., 2019) allows for optimized calculation, GPU support and enabling the use of automatic differentiation.

### 3.6 Adaptive collocation sampling

To allow for prioritized sampling of important locations regions, we optimize the positions of collocation points via gradient descent on the data of states history, projecting them back to the partition if the gradient step moves them outside to make sure there is a sufficient number of collocation point in each partition cell. We can calculate the gradient of each collocation point along each dimension from the continuity predictions and then use this gradient vector to optimize the position of the point by weighted sum as illustrated in Fig. 3.

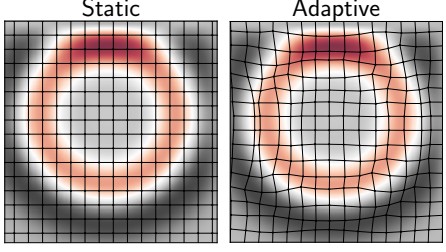

Figure 3: Adaptive collocation strategy: mesh points converge towards areas with higher information density.

We use the states at optimized positions adapted from history rollouts as the next rollouts input. By adapting the collocation points position our model is able to place greater emphasis in harder parts of the space to get a more accurate prediction.

### 3.7 Training strategy and loss function

To train the model, the collocation points at the initialized state are input to the MPNN to propagate rollouts on these points as shown in Fig. 4. Then two OSC methods allow for spatio–temporal continuous outputs, so that we can use any states in the prediction range as a target to train our model end-to-end. More specifically, given an input $\{\mathbf{X}, \mathbf{Y}^0\}$, the model outputs the simulated polynomial $\hat{u}(\mathbf{x}, t), \mathbf{x} \in \Omega, t \in [0, T]$ for a $T$ seconds prediction. By making use of higher resolution sample points along time and space in the train set as the target $(\mathbf{x}_i, y_i^{t_k})_{i=0,k=0}^{i=N_s,k=N_t}$, where $N_s$ is the number of target sample point and $N_t$ is the number of target sample time frames, we can calculate the reconstruction loss of sample points $L_s$. Moreover, we also calculate the reconstruction loss of predicted collocation points $L_c$. The complete loss is

$$L = \underbrace{\sum_{i=0}^{N_c}\sum_{k=0}^{N_k}\|y_i^{t_k} - \hat{y}_i^{t_k}\|^2}_{L_c \equiv \texttt{collocation points reconstruction}} + \underbrace{\sum_{i=0}^{N_s}\sum_{k=0}^{N_t}\|y_i^{t_k} - \hat{u}(\mathbf{x}_i, t_k)\|^2}_{L_s \equiv \texttt{sample points reconstruction}} \qquad (2)$$

where the $N_k$ is the number of MPNN rollout steps. The whole model is then trained end-to-end with automatic differentiation through the OSC.

## 4 EXPERIMENTS

### 4.1 DATASETS AND TRAINING

We test our GRAPHSPLINENETS on three settings of increasing challenge:

- **Heat Equation**: this PDE describes the behavior of heat through a domain and is characterized by diffusivity leading to stable solutions over time.
- **Damped Wave**: this system describes the propagation of waves through a medium characterized by their velocity; an additional damping term smooths their amplitude over time.

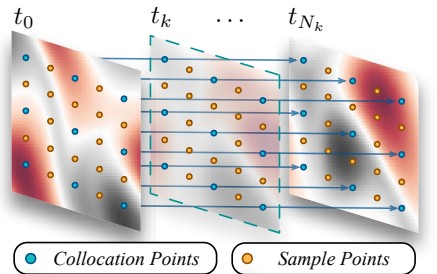

Figure 4: Data points used for model training.

- **Navier-Stokes**: these equations describe the behavior of 2D incompressible fluid flows; turbulence leading to chaoticity makes them a notoriously hard problem to solve.

All the models we test employ the same structure of encoder-processor-decoder for fair comparisons as well as the same amount of training data in each testing domain. While inputs of baseline models are directly all of the available data points, inputs of our OSC-based models are only an initialized $12 \times 12$ collocation point unitary mesh at the initial state and fewer in-time sample points.

### 4.2 EVALUATION METRICS AND BASELINES

We evaluated our model by calculating the average *mean square error* (MSE) of 1(s), 2(s), $\cdots$, 5(s) rollout predictions steps with respectively ground truth. We employ baselines relevant works in the field of discrete-step graph models for dynamical system predictions. *Graph convolution networks* (GCNs) (Kipf & Welling, 2016) and GCN with a hybrid *multilayer perceptron* (MLP) model are employed as baselines in the ablation study. We also compare our approach with one widely used baseline that employs linear interpolation for physics simulations allowing for continuous predictions in space, i.e., GEN (Alet et al., 2019). A similar setup considering the inherent graph structure and employing message passing neural networks in mesh space is employed by (Pfaff et al., 2021) (MPNN in our table). We employ the latter as the first building block for our GRAPHSPLINENETS.

### 4.3 QUANTITATIVE ANALYSIS

Empirical quantitative results on the three dataset are shown in Table 1. In the heat equation dataset, our approach reduces long range prediction errors by $64\%$ error while only using $20\%$ of the running time compared with the best baseline model. In the damped wave dataset, our approach reduces errors by $42\%$ with a $48\%$. In the Navier-Stokes dataset, our approach reduces $31\%$ long-range prediction errors while requiring $37\%$ less time to infer solutions compared to the strongest baseline.

### 4.4 ABLATION STUDY

We consider an ablation study on three datasets to demonstrate the effectiveness of our model components in our approach in multiple aspects. Quantitative results of ablation study models are shown in Table 1 [right]. The ablated models are:

- MPNN: base graph model with 3 message passing layers.

Table 1: Mean square error (MSE) propagation at different time stamps in seconds. Runtimes consider model inference for the full rollouts. Smaller is better (↓). Best in **bold**; second underlined.

| Dataset | Metric | | Baselines | | | | GRAPHSPLINENETS | | |
|---|---|---|---|---|---|---|---|---|---|
| | | | GCN | GCN+MLP | GEN | MPNN | MPNN+OSC(Post) | MPNN+OSC | MPNN+OSC+Adaptive |
| Heat Equation | MSE ($\times 10^{-3}$) | 1(s) | $0.52 \pm 0.09$ | $0.48 \pm 0.05$ | $\mathbf{0.23} \pm 0.02$ | $0.37 \pm 0.05$ | $0.38 \pm 0.04$ | $0.38 \pm 0.03$ | $\underline{0.28} \pm 0.03$ |
| | | 2(s) | $1.03 \pm 0.12$ | $0.87 \pm 0.08$ | $\underline{0.53} \pm 0.06$ | $0.60 \pm 0.06$ | $0.69 \pm 0.07$ | $0.56 \pm 0.05$ | $\mathbf{0.46} \pm 0.04$ |
| | | 3(s) | $2.58 \pm 0.22$ | $2.05 \pm 0.10$ | $1.53 \pm 0.09$ | $1.85 \pm 0.13$ | $1.28 \pm 0.09$ | $\underline{0.94} \pm 0.08$ | $\mathbf{0.87} \pm 0.09$ |
| | | 4(s) | $4.12 \pm 0.42$ | $3.87 \pm 0.23$ | $2.08 \pm 0.16$ | $2.68 \pm 0.21$ | $1.49 \pm 0.11$ | $\underline{1.02} \pm 0.10$ | $\mathbf{0.96} \pm 0.18$ |
| | | 5(s) | $6.87 \pm 1.00$ | $5.02 \pm 0.89$ | $2.92 \pm 0.23$ | $3.01 \pm 0.38$ | $1.68 \pm 0.18$ | $\underline{1.14} \pm 0.11$ | $\mathbf{1.07} \pm 0.28$ |
| | Rollout steps # | | 50 | 50 | 50 | 50 | 12 | 12 | 12 |
| | Runtime [s] | | $3.26 \pm 0.12$ | $3.02 \pm 0.10$ | $6.87 \pm 0.10$ | $6.99 \pm 0.12$ | $1.52 \pm 0.09$ | $\mathbf{1.38} \pm 0.10$ | $\underline{1.41} \pm 0.12$ |
| Damped Wave | MSE ($\times 10^{-1}$) | 1(s) | $1.61 \pm 0.11$ | $1.41 \pm 0.19$ | $\mathbf{0.71} \pm 0.08$ | $0.79 \pm 0.10$ | $0.81 \pm 0.09$ | $0.78 \pm 0.08$ | $\underline{0.74} \pm 0.09$ |
| | | 2(s) | $3.25 \pm 0.29$ | $2.85 \pm 0.27$ | $\mathbf{1.40} \pm 0.12$ | $1.60 \pm 0.15$ | $1.69 \pm 0.16$ | $1.59 \pm 0.15$ | $1.41 \pm 0.14$ |
| | | 3(s) | $5.12 \pm 0.48$ | $4.88 \pm 0.40$ | $2.98 \pm 0.28$ | $3.27 \pm 0.23$ | $2.57 \pm 0.18$ | $\underline{2.48} \pm 0.20$ | $\mathbf{2.28} \pm 0.24$ |
| | | 4(s) | $7.77 \pm 0.93$ | $6.01 \pm 0.82$ | $4.34 \pm 0.41$ | $5.27 \pm 0.41$ | $3.88 \pm 0.25$ | $\underline{3.41} \pm 0.22$ | $\mathbf{3.36} \pm 0.25$ |
| | | 5(s) | $10.5 \pm 1.65$ | $9.90 \pm 1.52$ | $6.49 \pm 0.62$ | $7.82 \pm 0.88$ | $4.98 \pm 0.29$ | $\underline{4.60} \pm 0.27$ | $\mathbf{4.51} \pm 0.31$ |
| | Rollout steps # | | 10 | 10 | 10 | 10 | 5 | 5 | 5 |
| | Runtime [s] | | $0.95 \pm 0.08$ | $0.82 \pm 0.07$ | $1.13 \pm 0.09$ | $1.38 \pm 0.10$ | $0.45 \pm 0.05$ | $\mathbf{0.39} \pm 0.04$ | $\underline{0.42} \pm 0.09$ |
| Navier Stokes | MSE ($\times 10^{-1}$) | 1(s) | $1.47 \pm 0.10$ | $1.22 \pm 0.11$ | $\mathbf{0.42} \pm 0.07$ | $0.66 \pm 0.10$ | $0.72 \pm 0.09$ | $0.70 \pm 0.08$ | $\underline{0.54} \pm 0.09$ |
| | | 2(s) | $2.01 \pm 0.21$ | $1.76 \pm 0.20$ | $\underline{0.98} \pm 0.10$ | $1.13 \pm 0.11$ | $1.20 \pm 0.11$ | $1.02 \pm 0.11$ | $\mathbf{0.80} \pm 0.14$ |
| | | 3(s) | $2.81 \pm 0.39$ | $2.45 \pm 0.36$ | $1.63 \pm 0.12$ | $1.64 \pm 0.24$ | $1.66 \pm 0.18$ | $\underline{1.44} \pm 0.20$ | $\mathbf{1.23} \pm 0.20$ |
| | | 4(s) | $3.51 \pm 0.64$ | $2.94 \pm 0.62$ | $2.38 \pm 0.16$ | $2.57 \pm 0.28$ | $1.98 \pm 0.24$ | $\underline{1.72} \pm 0.27$ | $\mathbf{1.50} \pm 0.29$ |
| | | 5(s) | $4.24 \pm 0.95$ | $3.91 \pm 0.99$ | $3.45 \pm 0.24$ | $3.66 \pm 0.33$ | $2.58 \pm 0.28$ | $\underline{2.21} \pm 0.27$ | $\mathbf{2.02} \pm 0.30$ |
| | Rollout steps # | | 10 | 10 | 10 | 10 | 5 | 5 | 5 |
| | Runtime [s] | | $0.91 \pm 0.08$ | $0.88 \pm 0.07$ | $1.01 \pm 0.09$ | $1.21 \pm 0.10$ | $0.51 \pm 0.05$ | $\mathbf{0.47} \pm 0.04$ | $\underline{0.49} \pm 0.09$ |

- MPNN+OSC(Post): model with 3 message passing layers and only post-processing with the OSC method, i.e., we firstly train a MPNN model, then we use the OSC method to collocate the prediction as a final result without end-to-end training.

- MPNN+OSC: MPNN with OSC-in-the-loop that allows for end-to-end training.

- MPNN+OSC+Adaptive: MPNN+OSC model that additionally employs our adaptive collocation point sampling strategy.

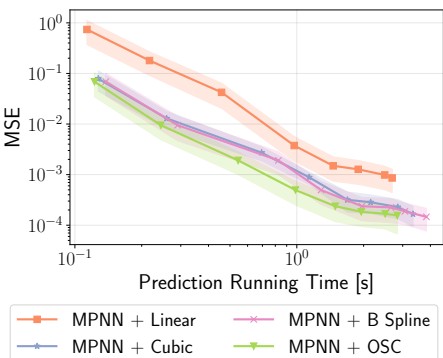

Figure 5: Pareto front of different interpolation and collocation points in terms of accuracy and speed with MPNNs clearly exhibits the advantages of the OSC method.

**Interpolation and collocation methods** We demonstrate the efficiency of the OSC method by comparing the combination of MPNN with different interpolation and collocation methods, including linear interpolation, cubic interpolation, and B-spline collocation methods. These models are implemented in the end-to-end training loop and we use these methods in both the time and space dimensions. Experiment results are shown in Fig. 5 where we measured the mean square error and the running time of 3 second rollouts predictions. We also test each method with a different number of collocation points, i.e. from $(2 \times 2)$ to $(16 \times 16)$ in Fig. 6. The model MPNN+OSC shows the best performance in obtaining the highest accuracy prediction among these approaches with shorter running time. Even though the linear interpolation can be faster than the OSC, it shows a considerable error n the prediction and does not satisfy basic assumptions such as Lipschitz continuity in space.

**Number of collocation points** We study the effect of the number of collocation points on the 3 second rollout prediction error by testing the MPNN, MPNN+OSC and MPNN+OSC+Adaptive models. The MPNN will always be directly trained with the whole domain data. We use different number of collocation points (from $(2 \times 2)$ to $(28 \times 28)$) into the MPNN process in the rest two models and then compare the output of the OSC with the whole domain to train. With the increase in number of collocation points, Fig. 6 shows that the MPNN+OSC and MPNN+OSC+Adaptive achieve significant improvements in prediction accuracy over the MPNN. The MPNN+OSC+Adaptive has a stable better performance compare with the MPNN+OSC and the improvement is larger when there are fewer collocation points. The reason is the with fewer collocation points, the MPNN+OSC has insufficient ability to learn the whole domain. With the help of the adaptive collocation point,

the MPNN+OSC+Adaptive can focus on hard-to-learning regions during training to obtain overall better predictions.

**COLROW solver and GPU acceleration** We show the effectiveness of the COLROW algorithm in accelerating the OSC speed by comparing the OSC method with one of the most commonly used algorithms for efficiently solving linear systems[2] and the OSC with the COLROW solver. Fig. 2 shows the experimental results 1000 solutions of ABD linear equations with the size of $256 \times 256$. Our package speeds up the OSC method by $32\%$. By making use of the GPU to speed up the OSC simulation, we can further lower the running time by $60\%$.

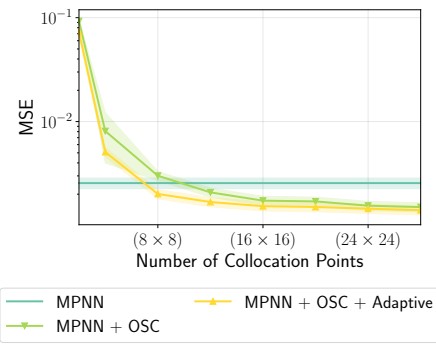

Figure 6: Number of collocation points and mean squared error.

**Number of rollout steps** We show the effectiveness of the OSC method in improving long-range prediction accuracy by comparing the MPNN and MPNN+OSC model. Fig. 7 shows the MPNN+OSC can keep stable in long-range rollouts compare with the MPNN. The reason is that with the OSC, we can use fewer neural network rollout steps to obtain a longer-range predictions, which avoids the error accumulation during the multi-step rollouts and implicitly learns for compensating integration residual errors. End-to-end learning lets the neural networks in MPNN+OSC learn the states between rollout steps, which will make the prediction stable and accurate.

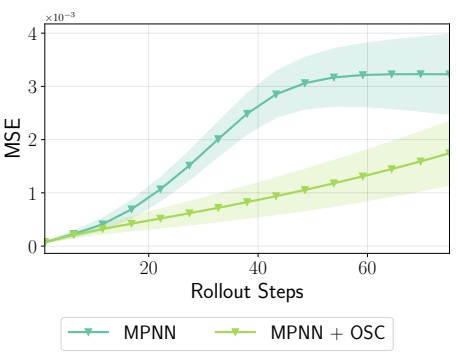

Figure 7: MPNN+OSC contains long rollout errors better than standard MPNN.

**Post processing vs end-to-end learning** We show the effectiveness of end-to-end learning architecture by comparing the MPNN+OSC(Post) and MPNN+OSC models. Table 1 show that MPNN+OSC has a more accurate prediction than MPNN+OSC(Post) by more than $8\%$ percent across datasets. This can be explained by the fact that, since the OSC is applied end-to-end, the error between MPNN prediction steps is backpropagated to the message passing layers, while in the post processing steps the model has no way of considering such error.

**Adaptive collocation point** We further show the effectiveness of the adaptive collocation strategy by comparing the MPNN+OSC and MPNN+OSC+Adaptive. Fig. 6 shows that the MPNN+OSC+Adaptive has a better performance than the MPNN+OSC in all collocation point setups. And Table 1 shows that MPNN+OSC+Adaptive has a more accurate prediction than MPNN+OSC, i.e. around $10\%$ improvement on long rollouts in the Navier-Stokes dataset. Adaptive collocation points encourage those points to move to the most dynamic regions in the domain, which is not only able to place greater emphasis on hard-to-learn parts in space, but can let the OSC method develop a better implicit representation of the domain.

## 4.5 QUALITATIVE ANALYSIS

We visualize the damped wave equation propagation and the Navier-Stokes evolution results of the GCN, GEN and MPNN+OSC(Our) in Fig. 8, Fig. 9 and Fig. 10. Our model has a smoother error distribution and more stable long-range prediction. Thanks to GRAPHSPLINENETS continuous predictions, we can simulate high resolutions without needing additional expensive model inference routines, while the other two models can only achieve lower resolution predictions. For long-range

---

[2]We used for our experiments `torch.linalg.solve`, which uses LU decomposition with partial pivoting and row interchanges. This is faster and numerically stable than matrix inversion; however, it has still a $O(n^3)$ time complexity.

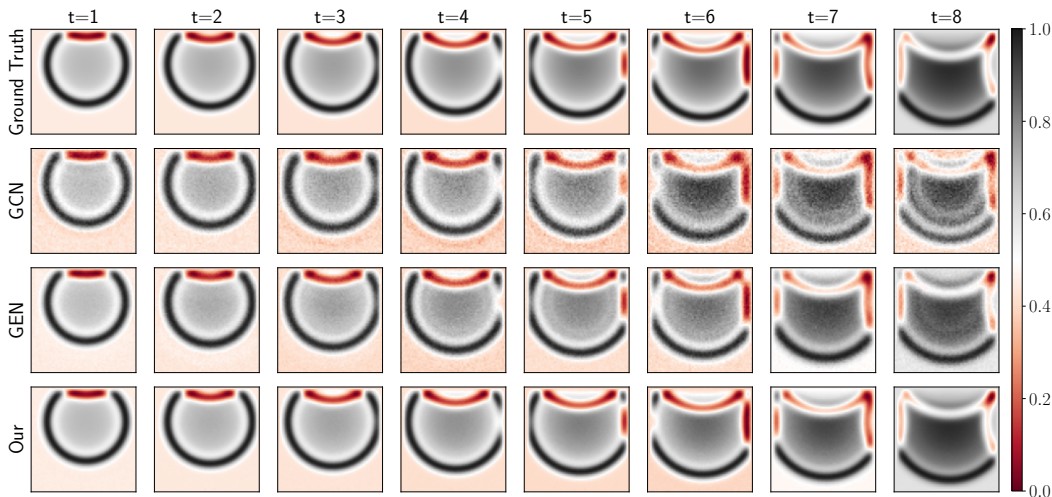

Figure 8: The visualization of results and the error on the wave dataset. Black dots in the collocation points figure are the position of sample points for our models training.

Figure 9: Wave dataset prediction results. GRAPHSPLINENETS manages to obtain more stable and smoother predictions compared to baselines.

predictions, while baselines visibly accumulate error, our model can lower the error with smoother and more accurate predictions in space and time.

## 5 CONCLUSION

We introduce GRAPHSPLINENETS, a novel method that aims at briding the gap between inherent discrete graph predictions in space and time and the continuous essence of natural processes. Our approach integrates the theory of Orthogonal Spline Collocation (OSC) methods to achieve space and time continuous simulations without the need for computationally expensive numerical routines. We introduce an effective adaptive collocation strategy to prioritize sampling of points in the space domain and implement the OSC end-to-end for achieving continuous predictions. We demonstrate how GRAPHSPLINENETS are robust in predicting processes which are characterized by several different PDEs arising directly from the differential equations. We believe this work represents an important step forward in a new direction in the research area at the intersection of deep learning and dynamical systems that aims at finding fast and accurate learned surrogate models.

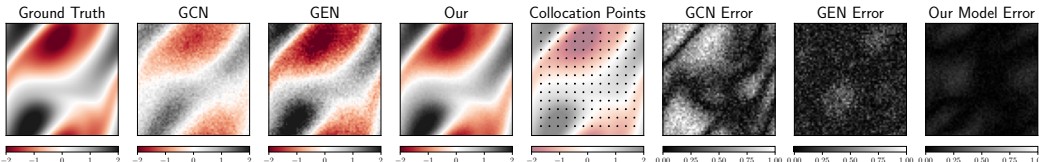

Figure 10: Qualitative results and errors on the Navier-Stokes dataset. Black dots in the collocation points figure are the positions of sample points for our models training.

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

# A  ADDITIONAL OSC MATERIAL

We further illustrate the OSC method by providing numerical examples in this section.

## A.1  1-D OSC EXAMPLE

For simplicity and without loss of generality, we consider the function domain as unit domain $[0, 1]$ and we set $N = 3, r = 2$, which means we will use a three-order three-piece function to simulate the 1-D ODE problem as shown in Equation. 2. We firstly choose the partition points as $x_i, i = 0, \cdots, 3, x_0 = 0, x_3 = 1$. The number of partition points is $N + 1 = 4$. Distance between partition points can be fixed or not fixed. Then we based on Gauss-Legendre quadrature rule choose collocation points. The number of collocation point within one partition is $r - 1 = 1$, so we have in total $N \times (r - 1) = 3$ collocation points $\xi_i, i = 0, \cdots, 3$.

After getting partition points and collocation points, we will construct the simulator. Here we have three partitions, in each partition, we assign a 2 order polynomial

$$a_{0,0} + a_{0,1}x + a_{0,2}x^2, x \in [x_0, x_1] \tag{3a}$$

$$a_{1,0} + a_{1,1}x + a_{1,2}x^2, x \in [x_1, x_2] \tag{3b}$$

$$a_{2,0} + a_{2,1}x + a_{2,2}x^2, x \in [x_2, x_3] \tag{3c}$$

Notice that these three polynomials should be $C^1$ continuous at the connecting points, i.e. partition points within the domain. For example, Equation Eq. (3a) and Equation Eq. (3b) should be continuous at $x_1$, then we can get two equations

$$\begin{cases} a_{0,0} + a_{0,1}x_1 + a_{0,2}x_1^2 & = a_{1,0} + a_{1,1}x_1 + a_{1,2}x_1^2 \\ 0 + a_{0,1} + 2a_{0,2}x_1 & = 0 + a_{1,1} + 2a_{1,2}x_1 \end{cases} \tag{4}$$

For boundary condition

$$\hat{u}(x) = \begin{cases} b_1, x = x_0 \\ b_2, x = x_3 \end{cases} \tag{5}$$

we can also get two equations

$$\begin{cases} a_{0,0} + 0 + 0 & = b_1 \\ a_{1,0} + a_{1,1} + a_{1,2} & = b_2 \end{cases} \tag{6}$$

Sum up equations we get so far. Firstly our undefined polynomials have $N \times (r+1) = 9$ parameters. The $C^1$ continuous condition will create $(N - 1) \times 2 = 4$ equations and the boundary condition will create 2 equations. Then we have $N \times (r - 1)$ collocation points. For each collocation point, we substitute it to polynomials to get an equation. For example, if the ODE is

$$\hat{u}(x) + \hat{u}'(x) = f(x), x \in [0, 1] \tag{7}$$

By substituting collocation point $\xi_0$ into the equation, we can get

$$\hat{u}(\xi_0) + \hat{u}'(\xi_0) = f(\xi_0)$$
$$\implies a_{0,0} + a_{0,1}\xi_0 + a_{0,2}\xi_0^2 + a_{0,1} + 2a_{0,2}\xi_0 = f(\xi_0) \tag{8}$$
$$\implies a_{0,0} + a_{0,1}(\xi_0 + 1) + a_{0,2}(\xi_0^2 + 2\xi_0) = f(\xi_0)$$

Now we can know that the number of equations can meet with the degree of freedom of polynomials

$$\underbrace{(r + 1) \times N}_{\text{Parameters}} = \underbrace{2}_{\text{Boundary}} + \underbrace{(N - 1) \times 2}_{C^1 \text{Continuous}} + \underbrace{N \times (r - 1)}_{\text{Collocation}} \tag{9}$$

At this example, generated equations will be constructed to an algebra problem $\mathbf{Aa} = \mathbf{f}$ where the weight matrix is an ABD matrix.

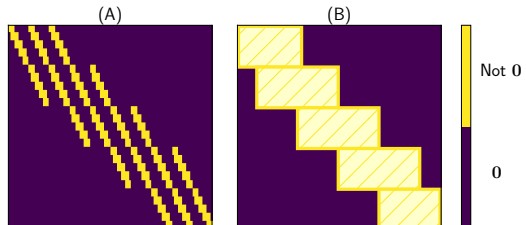

Figure 11: Visualization of an ABD matrix.

$$\mathbf{A} = \begin{bmatrix} 1 & 0 & 0 & 0 & 0 & 0 \\ 1 & \xi_0 + 1 & \xi_0^2 + 2\xi_0 & 0 & 0 & 0 \\ 1 & x_1 & x_1^2 & -1 & -x_1 & -x_1^2 \\ 0 & 1 & 2x_1 & 0 & -1 & -2x_1 \\ 0 & 0 & 0 & 1 & \xi_1 + 1 & \xi_1^2 + 2\xi_1 \\ 0 & 0 & 0 & 1 & 1 & 1 \end{bmatrix}, \tag{10a}$$

$$\mathbf{a} = \begin{bmatrix} a_{0,0} \\ a_{0,1} \\ a_{0,2} \\ a_{1,0} \\ a_{1,1} \\ a_{1,2} \end{bmatrix}, \mathbf{f} = \begin{bmatrix} b_1 \\ f(\xi_0) \\ 0 \\ 0 \\ f(\xi_1) \\ b_2 \end{bmatrix}. \tag{10b}$$

To solve this problem, we can get the simulation results.

### A.2 2–D OSC EXAMPLE

For simplicity and without loss of generality, we consider the function domain as unit domain $[0, 1] \times [0, 1]$ and we set $N_x = N_y = 2, r = 3$. Partition points and collocation points selection are similar with 1-D OSC method, we have $N^2 \times (r-1)^2 = 16$ collocation points in total. For simplicity, we note the partition points at two dimensions to be the same, i.e. $x_i, i = 0, 1, 2$. Unlike the 1–D OSC method, we choose Hermit bases to describe as the simulator, which keeps $C^1$ continuous. As a case, the base function at point $x_1$ would be

$$H_1(x) = f_1(x) + g_1(x)$$

$$f_1(x) = \begin{cases} \frac{(x-x_0)(x_1-x)^2}{(x_1-x_0)^2}, & x \in (x_0, x_1] \\ \frac{(x-x_2)(x-x_1)^2}{(x_2-x_1)^2}, & x \in (x_1, x_2] \end{cases} \tag{11}$$

$$g_1(x) = \begin{cases} +\frac{[(x_1-x_0)+2(x_1-x_0)](x-x_0)^2}{(x_1-x_0)^3}, & x \in (x_0, x_1] \\ +\frac{[(x_2-x_1)+2(x-x_1)](x_2-x)^2}{(x_2-x_1)^3}, & x \in (x_1, x_2] \end{cases}$$

We separately assign parameters to basis functions, i.e. $H_1(x) = a_{1,i}f_1(x) + b_{1,i}g_1(x)$ for $x$ variable in $[x_0, x_1] \times [y_{i-1}, y_i]$ partition. Then the polynomial in a partition is the multiple combinations of bases functions of two dimensions. For example, the polynomial in the partition $[x_0, x_1] \times [y_0, y_1]$ is

$$[a_{0,1}^x f_0(x) + b_{0,1}^x g_0(x) + a_{1,1}^x f_1(x) + b_{1,1}^x g_1(x)]$$
$$\times [a_{0,1}^y f_0(y) + b_{0,1}^y g_0(y) + a_{1,1}^y f_1(y) + b_{1,1}^y g_1(y)] \tag{12}$$

Now we consider the freedom degree of these polynomials. From definition, we have $2n(r-1)(n+1) = 24$ parameter. Consider boundary conditions, we have $24 - 4 \times N = 16$ parameters. The number is equal with collocation points $N^2 \times (r-1)^2$, which means we can get an algebra equation by substituting collocation points. Solving this equation we can get the simulator parameters.

We can similarly multiple basis functions and set parameters to the simulation result for the higher dimension OSC method. And then select partition points and collocation points by the same strategy with 2–D OSC method. The rest algebra equation generating and solving equations parts will not be different.

## A.3 SIMPLE NUMERICAL EXAMPLE

We set $N = 3, r = 3$ to simulate the problem

$$
\begin{cases}
u + u' = \sin(2\pi x) + 2\pi\cos(2\pi x) \\
u(0) = 0 \\
u(1) = 0
\end{cases}
\tag{13}
$$

we can get a simulation solution as following, with is visualized in Fig. 12.

$$
\hat{u}(x) = \begin{cases}
6.2x - 0.4x^2 - 31.4x^3, x \in [0, 1/3) \\
1.5 + 1.6x - 13.8x^2 + 9x^3, x \in [1/3, 2/3) \\
28.5 - 100x + 108.5x^2 - 37x^3, x \in [2/3, 1]
\end{cases}
\tag{14}
$$

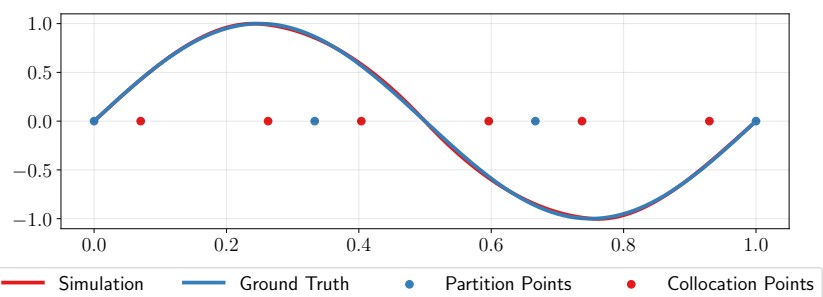

Figure 12: Visualization of an OSC solution.

## A.4 NUMERICAL ANALYSIS FOR INTERPOLATION AND COLLOCATION METHODS

We compared the OSC with linear, bilinear, 0–D cubic, and 2–D cubic interpolation methods on four types of problem: 1–D linear, 1–D non-linear, 2–D linear, and 2–D non-linear problems. At these experiments, we tested different simulator orders of the OSC method. For example, we set the order of the simulator to $4$ for 1–D linear problem and $2$ for 2–D linear problem. When the order of the simulator matches the polynomial order of the real solution, OSC can directly find the real solution. For non-linear problems, increasing the order of the simulator would be an ideal way to get lower loss. For example, we set the order of the simulator to $4$ for 1–D non-linear problem and $5$ for 2–D non-linear problem. Thanks to the efficient calculation of OSC, even though we use higher–order polynomials to simulate, we use less running time to get results.

Table 2: Error of OSC and four interpolation methods on different PDEs problems: $u(x) = x^4 - 2x^3 + 1.16x^2 - 0.16x$ (1-D linear), $u(x) = \sin(3\pi x)$ (1-D non-linear), $u(x, y) = x^2 y^2 - x^2 y - xy^2 + xy$ (2-D linear), $u(x, y) = \sin(3\pi x)\sin(3\pi y)$ (2-D non-linear).

| MODEL | 1-D LINEAR | 1-D NON-LINEAR | 2-D LINEAR | 2-D NON-LINEAR |
|---|---|---|---|---|
| NEAREST INTERPOLATION | $2.3670 \times 10^{-6}$ | $1.7558 \times 10^{-2}$ | $1.9882 \times 10^{-3}$ | $3.8695 \times 10^{-2}$ |
| LINEAR INTERPOLATION | $1.8928 \times 10^{-7}$ | $8.7731 \times 10^{-4}$ | $3.4317 \times 10^{-4}$ | $1.1934 \times 10^{-2}$ |
| CUBIC INTERPOLATION | $3.5232 \times 10^{-12}$ | $2.2654 \times 10^{-7}$ | $2.9117 \times 10^{-4}$ | $4.5441 \times 10^{-3}$ |
| OSC | $\mathbf{3.4153 \times 10^{-31}}$ | $\mathbf{4.1948 \times 10^{-8}}$ | $\mathbf{1.7239 \times 10^{-32}}$ | $\mathbf{3.4462 \times 10^{-5}}$ |

# B ADDITIONAL EXPERIMENTAL DETAILS

## B.1 HEAT EQUATION

The heat equation describes the diffusive process of heat conveyance and can be defined by

$$\frac{\partial u}{\partial t} = \Delta u \tag{15}$$

where $u$ denotes the solution to the equation and $\Delta$ is the Laplacian operator over the domain. In a $n$-dimensional space, it can be written as:

$$\Delta u = \sum_{i=1}^{n} \frac{\partial^2 u}{\partial x_i^2} \tag{16}$$

**Dataset generation** We employ FEniCS (Logg et al., 2012) to generate a mesh from the domain and solve the heat equation on these points. The mesh is then used by the graph neural network for training.

### B.2 WAVE EQUATION

The damped wave equation can be defined by

$$\frac{\partial^2 w}{\partial t^2} + k\frac{\partial w}{\partial t} - c^2 \Delta w = 0$$

where $c$ is the wave speed and $k$ is the damping coefficient. The state is $X = (w, \frac{\partial w}{\partial t})$.

**Data generation** We consider a spatial domain $\Omega$ represented as a $64 \times 64$ grid and discretize the Laplacian operator. $\Delta$ is implemented using a $5 \times 5$ discrete Laplace operator in simulation; null Neumann boundary condition are imposed for generation. We set $c = 330$ and $k = 50$ similarly to the original implementation in Yin et al. (2021).

### B.3 NAVIER-STOKES

The Navier-Stokes equations describe the dynamics of incompressible flows with a 2-dimensional PDE. They can be described in vorticity form as:

$$\begin{aligned} \frac{\partial w}{\partial t} &= -v\nabla w + \nu\Delta w + f \\ \nabla v &= 0 \\ w &= \nabla \times v \end{aligned} \tag{17}$$

where $v$ is the velocity field and $w$ is the vorticity, $\nu$ is the viscosity and $f$ is a forcing term. The domain is subject to periodic boundary conditions.

**Data generation** We generate trajectories with a temporal resolution of $\Delta t = 1$ and a time horizon of $t = 10$. We use similar settings as in Yin et al. (2021) and Kirchmeyer et al. (2022): the space is discretized on a $64 \times 64$ grid and we set $f(x, y) = 0.1(\sin(2\pi(x + y)) + \cos(2\pi(x + y)))$, where $x, y$ are coordinates on the discretized domain. We use a viscosity value $\nu = 10^{-3}$.

### B.4 MODELS AND IMPLEMENTION DETAILS

For all experiments, a batch size of 32 was used and the models were trained for up to 5000 epochs with early stopping. We used the Adam optimizer (Kingma & Ba, 2014) with an initial learning rate of 0.001 and a step scheduler with a 0.85 decay rate every 500 epochs. For all datasets, we used the split of $5 : 1 : 1$ for training, validating and testing for fair comparison.

Specific details of model components are introduced below:

- *MPNN encoder*: a three-layer MLP with hidden size$= 64$.

- *MPNN processor*: in total 3 processors, each with three-layer MLP with hidden size$= 64$.

- *MPNN decoder*: a three-layer MLP with hidden size$= 64$.

All MLPs have ReLU : $x \rightarrow \max(0, x)$ nonlinearities between layers.

Specific details of applying the OSC method are introduced as follows:

- *Time–oriented OSC*: polynomial order= 3, number of collocation points in one partition= 2.
- *Space–oriented OSC*: x dimension polynomial order= 3, polynomial order in the y dimension= 3, number of collocation point in one partition on x dimension= 2; and number of collocation point in one partition on y dimension= 2.

## B.5 Hardware and Software

Experiments were carried out on a machine equipped with an INTEL CORE I9 7900X CPU with 20 threads and a NVIDIA RTX A5000 graphic card with 24 GB of VRAM. Software–wise, we used FEniCS (Logg et al., 2012) for Finite Element simulations for the heat equation experiments and PyTorch (Paszke et al., 2019) for simulating the damped wave and Navier-Stokes equations. We employed the Deep Graph Library (DGL) (Wang et al., 2020) for graph neural networks and the PyTorch Lightning library (Falcon et al., 2019) for training.

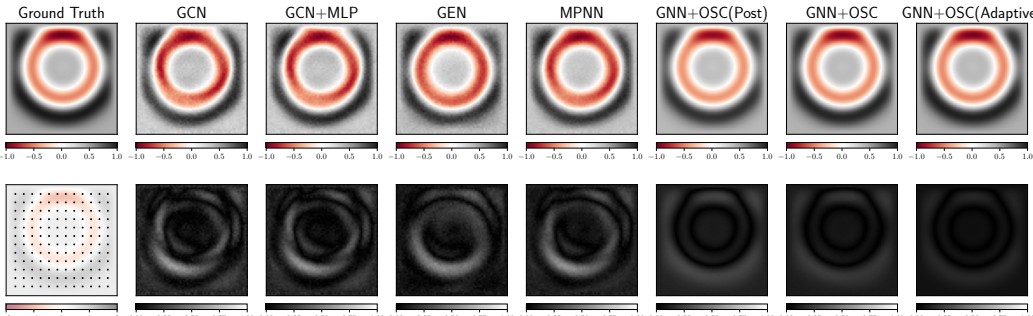

Figure 13: Visualization of the wave dataset.

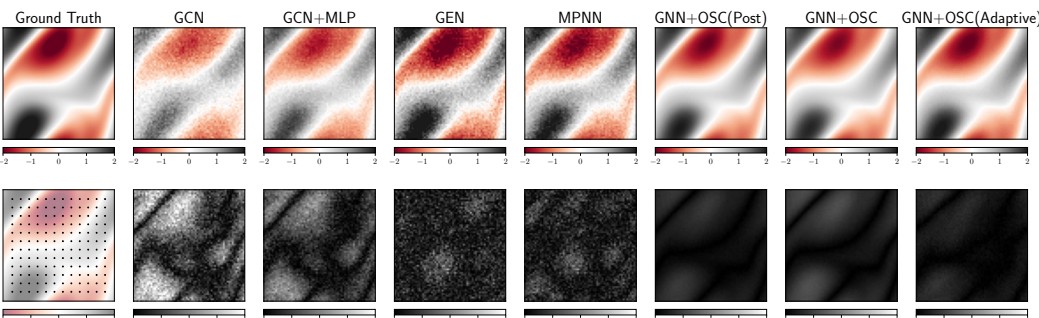

Figure 14: Visualization of the Navier Stokes dataset.

