# OpenReview forum: "Graph Spline Networks for Efficient Continuous Simulation of Dynamical Systems"
_ICLR.cc/2023/Conference — Submitted to ICLR 2023_

### Official Review · Reviewer_5HfZ · 2022-10-22

**Confidence:** 3
**Correctness:** 2
**Technical Novelty And Significance:** 2
**Empirical Novelty And Significance:** 2
**Recommendation:** 3

**Clarity, Quality, Novelty And Reproducibility:**

**Clarity** Seems OK overall. The OSC motivating examples in the supplemental material are helpful.

A minor note: the paper’s writing is not very well polished. There are quite a few typos and many small grammatical errors, e.g.,
FORTAN => FORTRAN
?? at the end of Sec. 3.5.
resul => result
2 order polynomial => 2nd-order polynomial
These typos did not affect my score, though.

**Quality and novelty** Please see my comments in “Strength and Weaknesses” before.

**Reproducibility** The technical method itself seems pretty easy to implement. The experiments may depend on some careful choices of hyperparameters, e.g., the number of splines or collocation points, the network depth, etc., but I don’t expect reproducing the experiments to be challenging. Of course, it would be much easier if the paper could release code.


**Strength And Weaknesses:**

The paper is a respectable submission, but I am not sure I am fully convinced by the proposed technical method. I have listed my questions below, and it would be good if the authors could address them during the rebuttal:

The whole method reminds me of the Physics-Informed Neural Network (PINN) [Raissi et al.], which also attempts to reconstruct a continuous spatial-temporal solution to PDE problems. The related work section seems to overlook PINN, but I think it deserves a more careful discussion to clarify the difference between this paper and PINNs.

Actually, I am not sure using OSC to construct continuous spatial-temporal solutions is as significant as the paper claims. With the nodal prediction from graph networks only, one can still generate continuous solutions with standard interpolation schemes (e.g., using the basis functions in the Galerkin method from FEM). I don’t think it is super challenging to obtain a continuous spatial-temporal solution once we have the finite-dimensional discretization, so I find it hard to see the necessity of fusing OSC into a learning model in the first place.

I am also not sure I agree with a few statements that this paper makes about solving linear systems of equations:
- “...with the most widely used linear equation solver (i.e., use the inverse matrix to solve the equation)”. Quite oppositely, directly inverting a matrix is perhaps the last thing people should do to solve a large linear system of equations.
- The time complexity analysis in Sec. 3.5 is also questionable to me: The ABD matrix seems quite sparse, so the time complexity for solving dense linear systems of equations does really apply here.
- I am also not sure COLROW is the state-of-the-art solver here since it was published nearly 40 years ago. Would it make more sense to consider more recent methods and packages, e.g., SuiteSparse, Eigen, or AMGCL? On a related note, if the ABD matrix is SPD, more dedicated solvers (e.g., Pardiso or CHOLMOD) can apply.


**Summary Of The Paper:**

This paper combines graph network simulators with orthogonal spline collocation (OSC) methods to model complex dynamic systems at continuous spatial and temporal locations. The technical method first applies standard graph network models to predict a dynamic system’s state at discrete spatial locations and time steps. Next, it applies OSC to augment the discrete results into continuous solutions. The whole model is end-to-end differentiable so that standard optimization algorithms can apply. The paper presents evaluation experiments on a few common PDE problems, e.g., heat equations and Navier-Stokes equations.

**Summary Of The Review:**

My major concern is mostly about the technical approach itself. While I appreciate the paper’s effort, I don’t think the proposed approach is significant or novel enough to reach the ICLR standard. However, I am open to different opinions from the authors or other reviewers.

---

### Official Review · Reviewer_ESAs · 2022-10-24

**Confidence:** 3
**Correctness:** 4
**Technical Novelty And Significance:** 3
**Empirical Novelty And Significance:** 2
**Recommendation:** 5

**Clarity, Quality, Novelty And Reproducibility:**

The paper is very clearly written, and cites relevant prior work. The used techniques are introduced in the text at an appropriate depth, with more details made available in the appendix, where the architectural details and hyperparameters of the models can also be found. The authors do not provide the code for their implementation, and reference provided for the "implementation details" is broken in the submitted version of the paper.

The proposed method is a novel combination of preexisting techniques (MPNNs + OSC), made more powerful by an end-to-end differentiable and GPU-accelerated implementation, allowing for colocation point optimization.

Suggestions:
- It would be interesting to see illustrations of MPNN (without OSC) and the various OSC-related ablations in Figs. 8-10. If space is a limitation, I think it would still be helpful to have them in the appendix.


**Strength And Weaknesses:**

The strengths of the paper include the experimental results showing the utility of the proposed method, and the inclusion of ablations illustrating the impact of OSC as a post-processing step, or when trained end-to-end with and without adaptive colocation points. It is also very helpful that the authors include error images in Figs 8 and 10.

A limitation is that all the presented testcases are ones with relatively simple structure and dynamics. It would be interesting how the method performs in a more complex situation. For instance, in the 2D NSE case, one could consider analyzing a Karman vortex street or unstable flow at a higher Reynolds numbers. There is also little novelty on the ML front -- the proposed method is more of a (trainable) postprocessing step for GNN-based learned surrogate models.


**Summary Of The Paper:**

The paper proposes to combine message-passing NNs with the orthogonal spline collocation (OSC) to build a method capable of predicting PDE solutions that are continuous in time and space and which can be efficiently evaluated to generate a value at arbitrary points in space and time within the simulated domain. The method is tested on the heat equation, damped wave, and 2d Navier-Stokes (incompressible fluid flow), and shown to be more efficient than classical interpolation schemes, to produce smoother solutions than baselines, and generate more stable long-range predictions.


**Summary Of The Review:**

The paper introduces an interesting combination of existing techniques, with experiments illustrating gains in accuracy and performance in a number of simple settings. This is interesting, but would be strengthened by more extensive experiments, or by more focus on reproducibility (ideally in the form of open source code).

---

### Official Review · Reviewer_mydW · 2022-10-25

**Confidence:** 3
**Correctness:** 3
**Technical Novelty And Significance:** 3
**Empirical Novelty And Significance:** 3
**Recommendation:** 3

**Clarity, Quality, Novelty And Reproducibility:**

I feel the paper can provide a clearer story and provide better experiment details

**Strength And Weaknesses:**

Strength
The physics application is quite interesting
Combining deel learning with physics are interesting




Weakness


1 experiment designs:
Heat Equation:  Damped Wave:  Navier-Stokes are simulated? not real world data?
simulated data will be much easier to fit.

2 motivation and clarity
not fully board with the motivation of Time–oriented OSC/Space–oriented OSC:  Is Time–oriented OSC/Space–oriented OSC for introducing continuous solution on mesh grid data?

I am not fully on board on why using GNN is helpful, because mesh-grid data?

3 evaluation
Table 1: existing methods seem not for this tasks specific (i.e. GNN in general), so performance on this tasks may not be good.
compare with methods that  without or without  “space and time continuous simulations” are more relevant. GEN is one of the example.
 Table is shows larger gain on 5s versus 1s, are there some discussion on this?


**Summary Of The Paper:**

The aurhors provide GRAPHSPLINENETS, a novel deep learning approach to speed up simulation of physical systems with spatio-temporal continuous outputs by exploiting the synergy between graph neural networks (GNN) and orthogonal spline collocation (OSC).

Two differentiable OSC (time-oriented OSC and spatial-oriented OSC) are applied to bridge the gap between discrete GNN outputs and generate continuous solutions at any location in space and time without explicit prior knowledge of underlying differential equations.

**Summary Of The Review:**

I have some concerns on "experiment design", "motivations", "evaluations"

---

### Decision · Program_Chairs · 2023-01-20

**Decision:**

Reject

**Justification For Why Not Higher Score:**

All reviewers are concerned with multiple aspects of this paper, including experimental comparisons and results, and technical methods and motivations, etc. Given these concerns, a reject is recommended.

**Justification For Why Not Lower Score:**

NA

**Metareview: Summary, Strengths And Weaknesses:**

This paper proposes to study physical simulations by combining graph neural nets and spline networks. All reviewers are concerned with multiple aspects of this paper, including experimental comparisons and results, and technical methods and motivations, etc. Given these concerns, a reject is recommended.